

# Effects of plant growth-promoting rhizobacteria on co-inoculation with *Bradyrhizobium* in soybean crop: a meta-analysis of studies from 1987 to 2018

Douglas M. Zeffa[1], Lucas H. Fantin[2], Alessandra Koltun[1], André L.M. de Oliveira[3], Maria P.B.A. Nunes[2], Marcelo G. Canteri[2] and Leandro S.A. Gonçalves[2]

[1] Department of Agronomy, Universidade Estadual de Maringá, Maringá, Paraná, Brazil
[2] Department of Agronomy, Universidade Estadual de Londrina, Londrina, Paraná, Brazil
[3] Department of Biochemistry and Biotechnology, Universidade Estadual de Londrina, Londrina, Paraná, Brazil

## ABSTRACT

**Background**. The co-inoculation of soybean with *Bradyrhizobium* and other plant growth-promoting rhizobacteria (PGPR) is considered a promising technology. However, there has been little quantitative analysis of the effects of this technique on yield variables. In this context, the present study aiming to provide a quantification of the effects of the co-inoculation of *Bradyrhizobium* and PGPR on the soybean crop using a meta-analysis approach.

**Methods**. A total of 42 published articles were examined, all of which considered the effects of co-inoculation of PGPR and *Bradyrhizobium* on the number of nodules, nodule biomass, root biomass, shoot biomass, shoot nitrogen content, and grain yield of soybean. We also determined whether the genus of the PGPR used as co-inoculant, as well as the experimental conditions, altered the effect size of the PGPR.

**Results**. The co-inoculation technology resulted in a significant increase in nodule number (11.40%), nodule biomass (6.47%), root biomass (12.84%), and shoot biomass (6.53%). Despite these positive results, no significant increase was observed in shoot nitrogen content and grain yield. The response of the co-inoculation varied according to the PGPR genus used as co-inoculant, as well as with the experimental conditions. In general, the genera *Azospirillum*, *Bacillus*, and *Pseudomonas* were more effective than *Serratia*. Overall, the observed increments were more pronounced under pot than that of field conditions. Collectively, this study summarize that co-inoculation improves plant development and increases nodulation, which may be important in overcoming nutritional limitations and potential stresses during the plant growth cycle, even though significant increases in grain yield have not been evidenced by this data meta-analysis.

Corresponding author
Leandro S.A. Gonçalves,
leandrosag@uel.br

## INTRODUCTION

The soybean crop (*Glycine max* (L.) Merrill) is one of the main commodities in the world, mainly for its high protein and oil contents, favoring its use in several areas of the agroindustry (*Hart, 2017*; *Nguyen, 2018*). In countries such as Brazil and Argentina, some of the world's leading producers, soybean is a highly profitable crop for farmers, since its nitrogen (N) requirements are fully met by biological nitrogen fixation (BNF) (*Hungria et al., 2005*). In BNF, the soybean establishes a symbiotic relationship with rhizobia, providing photoassimilates in exchange for biologically active N (*Hungria, Menna & Delamuta, 2015*; *Gresshoff, 2018*). These microorganisms usually inhabit the plant root system, where they colonize and grow endophytically, producing the enzyme complex nitrogenase, which allows them to convert atmospheric nitrogen ($N_2$) to ammonia and its further incorporation into biomolecules in several forms of organic N (*Hungria et al., 2006*; *Oldroyd, 2013*; *Hungria, Nogueira & Araujo, 2013*).

The genus *Bradyrhizobium* (*Jordan, 1982*) is considered the main rhizobial genus that establishes a symbiotic association with soybean (*Hungria, Nogueira & Araujo, 2015*; *Sugiyama et al., 2015*; *Schmidt, Messmer & Wilbois, 2015*). According to List of Prokaryotic Names with Standing in Nomenclature (*LPSN, 2019*), 41 species of *Bradyrhizobium* have already been described, with the species *B. elkanii*, *B. japonicum*, and *B. diazoefficiens* being the most used in commercial inoculants (*Siqueira et al., 2014*; *Schmidt, Messmer & Wilbois, 2015*; *Delamuta et al., 2017*). The *Bradyrhizobium*-soybean symbiosis is considered one of the most important natural relations exploited by the agricultural activity, since these bacteria can lead to grain yield increase and, consequently, eliminate or reduce the dependence on inorganic N fertilizers in crop cultivation (*Chang, Lee & Hungria, 2015*; *Hungria, Marco & Ricardo, 2015*; *Collino et al., 2015*).

In addition to the use of rhizobia, another strategy that has been employed to increase soybean productivity is the co-inoculation of *Bradyrhizobium* with other genera of plant growth-promoting rhizobacteria (PGPR), such as *Azospirillum* (*Hungria, Marco & Ricardo, 2015*; *Zuffo et al., 2016*), *Bacillus* (*Mishra et al., 2009*; *Tonelli, Magallanes-Noguera & Fabra, 2017*), *Pseudomonas* (*Egamberdieva, Jabborova & Berg, 2016*; *Pawar et al., 2018*), and *Serratia* (*Bai, 2002*; *Pan, Vessey & Smith, 2002*). These microorganisms act as promoters of plant growth via the production of amino acids, indole acetic acid (IAA), gibberellins, and other polyamines, improving root growth and, consequently, increasing water and nutrient absorption by the plants and generating rhizobia-soybean interaction sites (*Schmidt, Messmer & Wilbois, 2015*; *Yadav et al., 2017*). Among other benefits, PGPR are also able to solubilize phosphates, produce siderophores, fix $N_2$, and mitigate biotic and abiotic stresses (*Ahemad & Kibret, 2014*; *Olanrewaju, Glick & Babalola, 2017*). In the sense, the co-inoculation of microorganisms with different functions can be considered an economically viable and environmentally sustainable strategy to improve plant performance (*Muthukumar & Udaiyan, 2018*; *Yan, Zhu & Yang, 2018*).

Although it is considered a promising technology, the co-inoculation of soybean has shown contrasting results (*Schmidt, Messmer & Wilbois, 2015*). *Hungria, Nogueira & Araujo (2013)* investigating the effects of co-inoculation of soybean seeds with *B. japonicum*

and *A. brasilense*, observed an average increase of 420 kg ha$^{-1}$ (16.1%) compared to the control treatment inoculated only with *B. japonicum*. Conversely, *Zuffo et al. (2016)* reported no significant differences in grain yield between inoculated (*B. japonicum*) and co-inoculated (*B. japonicum* + *A. brasilense*) treatments for six soybean cultivars. Nevertheless, *Atieno et al. (2012)* observed that co-inoculation of *B. japonicum* and *B. subtilis* increased traits related to soybean nodulation and biomass. Therefore, what is not yet clear is the impact of co-inoculation on soybean grain yield. In view of this, the statistical technique known as meta-analysis may be a powerful tool to determine the real effects of the co-inoculation of PGPR and *Bradyrhizobium* on soybean cultivation, since this technique allows the quantitative combination of results from different studies, leading to a synthesis of results with high power and reliability. Therefore, the objective of this study was to investigate and solve the inconsistency of results using a meta-analysis.

## MATERIAL & METHODS

### Bibliographic research and data collection

Figure 1 shows the search strategy for the review presented according to the PRISMA reporting guidelines (*Liberati et al., 2009*). Data were collected from articles published in scientific journals, which were obtained by a systematic literature review using the Web of Science® and Google Scholar® databases. The search strategy "soybean AND (co-inoculation OR PGPR)" was applied in both databases in February 2018 by two independent reviewers (DMZ and LHF). Discussion between the two reviewers resolved any differences. If no consensus could be reached, another reviewer (LSAG) resolved the conflict. After screening relevant titles and filtering out duplicates, 79 articles were reviewed. The final article number was then reduced to 42 based on the following criteria: (i) articles written in English, Spanish, or Portuguese; (ii) studies that presented a measure of variance: coefficient of variation (CV), mean square residual (MSR), standard error of the mean (SE), or standard deviation of the mean (SD); (iii) studies showing the number of nodules, nodule biomass, shoot biomass, root biomass, shoot N content, and/or grain yield traits; and (iv) studies comparing inoculated treatments (*Bradyrhizobium*) × co-inoculated (*Bradyrhizobium* + PGPR). Interaction data with biotic or abiotic stresses were not extracted from articles.

Nodule, root, and shoot biomass were generally presented as dry biomass; however, in some cases, the values of fresh biomass were used when they were the only type of measure available. For the variable shoot N content, protein content was also used as an indirect source. The means and the measures of variance were extracted from the article tables, when provided. For figures, we extracted data using the ImageJ 1.5 software (*Pérez & Pascau, 2013*). Bar graphs that contained variance without specification were considered as SD.

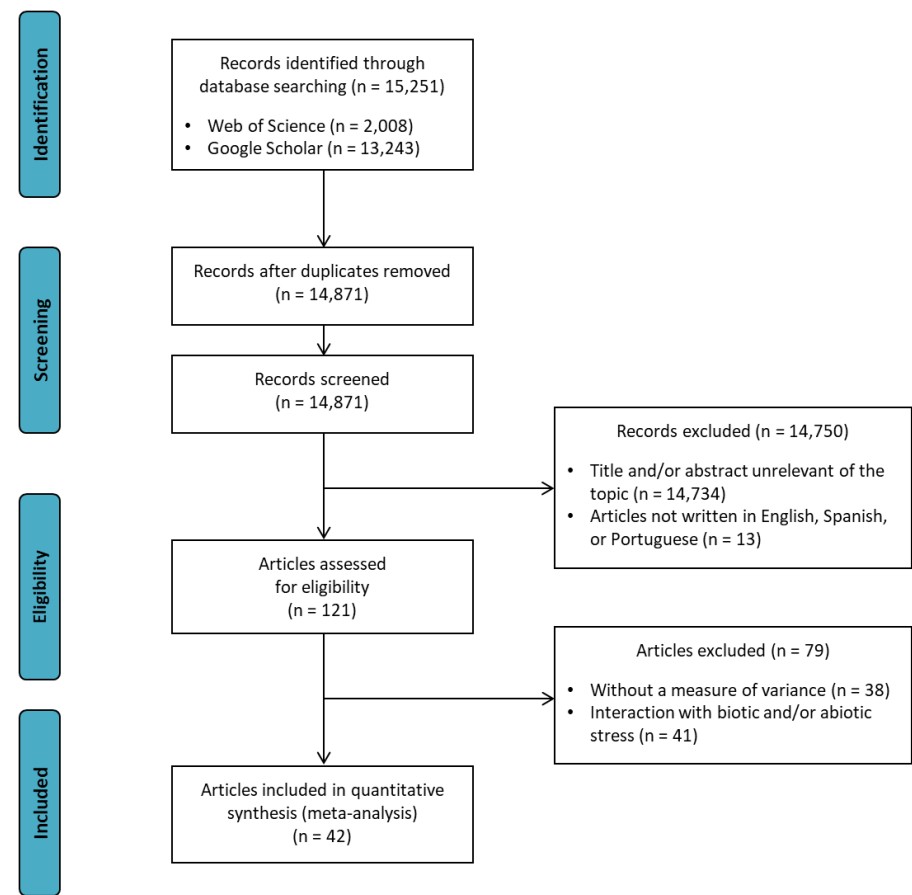

**Figure 1** Preferred reporting items for systematic reviews and meta-analyses (PRISMA) flow diagram for the meta-analysis.

## Effect size and moderator variables

Estimates of the effects of the PGPR on the evaluated traits were obtained using the natural logarithmic response ratio (ln R) as effect size (*Hedges, Gurevitch & Curtis, 1999*):

$$\ln R = \ln\left(\frac{Ti}{Tc}\right)$$

in which $Ti$ is the mean of the co-inoculated treatment (*Bradyrhizobium* + PGPR) and $Tc$ is the mean of the control treatment (*Bradyrhizobium*). The rate of the response is useful when different units are reported in the studies, while logarithmic transformation is necessary to properly balance the treatments of positive and negative effects to maintain symmetry within the analysis (*Cooper, Hedges & Valentine, 2009*). Thus, values above zero indicate an increase in the variable induced by PGPR, while values below zero reflect a reduction, and a value that equals zero means absence of the effect of PGPR. In addition, the ln R can be easily transformed into a percentage response (%R), using the following formula:

$$\%R = 100 \times [\exp.(\ln R) - 1]$$

Experimental conditions (field or pot) and PGPR genera used in co-inoculation were used as moderator variables in the present study, since they may influence the response of soybean to the effects of co-inoculation. Moderator variables were selected based on the criterion of a minimum of 15 observations in at least two scientific articles. The moderator variables were tested even when the evaluated trait presented no significant value, since the positive results may have been diluted in the general effect.

## Meta-analysis

Prior to the construction of the meta-analysis models, data heterogeneity was verified by the $Q$ (*Cochran, 1954*) and $I^2$ (*Higgins & Thompson, 2002*) tests to determine the use of fixed or random/mixed-effects model approaches. The synthesis produced by the meta-analysis is balanced according to the weight of each of the studies, so that they can contribute individually to the meta-analytic result. In this study, the inverse variance method (*Hedges, Gurevitch & Curtis, 1999*) was used to assign the weights:

$$Wi = \frac{1}{Vi}$$

in which $Wi$ represents the weight assigned to the $i$-th study and $Vi$ is the variance of the $i$-th study. Thus, the lower the study variance, the greater its contribution to the synthesis generated.

The estimates produced by the meta-analysis and their respective 95% confidence intervals (95% CI) were presented in forest plot graphs. Therefore, the mean effect size was considered significant when its 95% CI did not overlap with zero. Statistical analyses were performed in the software R (https://r-project.org), using the meta (*Schwarzer, 2007*), metafor (*Viechtbauer, 2010*), and ggplot2 (*Wickham, 2016*) packages.

## RESULTS

### Metadata

Metadata was obtained from 42 published articles from 13 countries between 1987 and 2018 (Fig. 2A; Table S1). A total of 976 observations ($n$) were obtained from an aggregate of 74 trials, where each observation included a co-inoculated treatment (PGPR + *Bradyrhizobium*) and a control treatment (*Bradyrhizobium*) for the number of nodules ($n = 278$), nodule biomass ($n = 228$), shoot N content ($n = 88$), and grain yield ($n = 78$). Among the observations, 53% ($n = 525$) were obtained in pots and 47% ($n = 451$) under field conditions (Fig. 2B). Except for grain yield, reported only under field conditions, all other traits were observed under pot and field conditions. A total of 16 different genera of PGPR were used as co-inoculants (Fig. 2C).

Heterogeneity on the full dataset was highly significant by the Cochran test ($Q = 29822.77$, $df = 975$, $p < 0.0001$). The $I^2$ statistic also indicated high heterogeneity, which showed a magnitude of 96.40%. Due to the great heterogeneity of the observations, the meta-analysis was performed using random-effects models. Likewise, significant heterogeneity ($p < 0.0001$) was observed for the six evaluated traits grouped by the moderator variables, suggesting the use of mixed-effects models, in which we evaluated

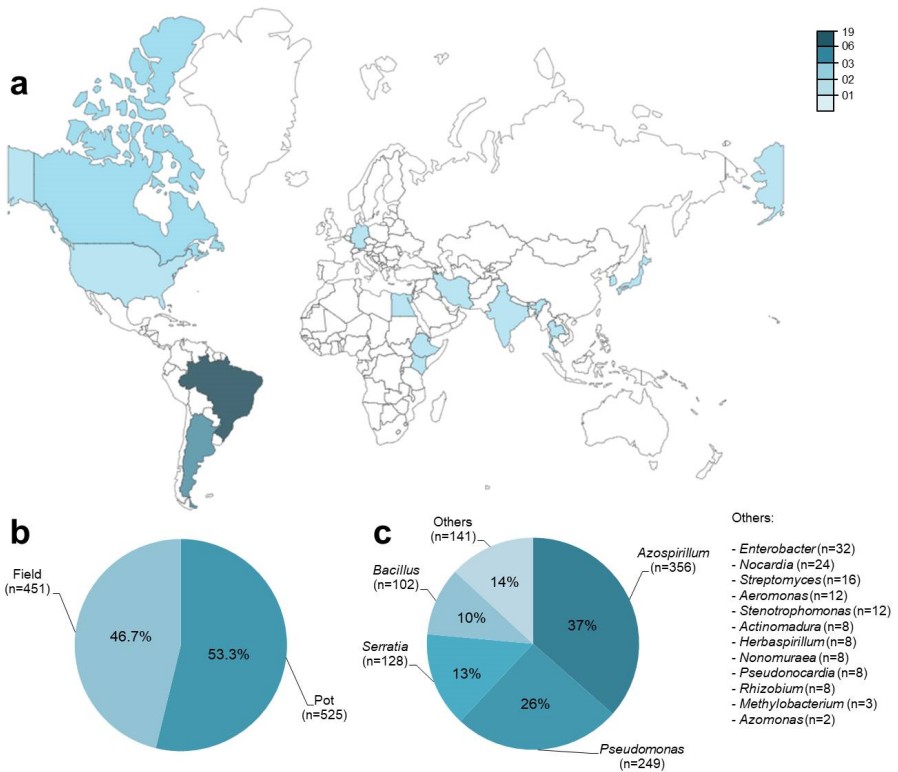

**Figure 2** General data information (*n* = 976) obtained from 42 studies used in the meta-analysis, according to (A) location of the experiments, (B) experimental conditions and (C) genera of PGPR used as co-inoculants.

the moderator variables as random effect covariates and the observations as fixed effects (*Cooper, Hedges & Valentine, 2009*).

## General effect of co-inoculation

The co-inoculation of soybean with PGPR showed a positive and significant effect on the number of nodules (11.40%, 95% CI [7.06 –15.93%]), nodule biomass (6.47%, 95% CI [0.59–12.70%]), root biomass (12.84%, 95% CI [3.64–22.85%]), and shoot biomass (6.53%, 95% CI [3.34–9.82%]) (Fig. 3). However, there was no increase in grain yield and shoot N content associated with co-inoculation, since their 95% CI overlapped with zero.

## Effects of the moderator variables

The effects of the moderator variables on the number of nodules are shown in Fig. 4. Regarding the experimental conditions, the tests conducted under field and pot conditions showed significant effects of 8.55% (95% CI [3.09–14.29%]) and 12.84% (95% CI [7.38–20.12%]), respectively, on the evaluated traits (Fig. 4A). Both effect sizes can be considered similar, since the 95% CI overlapped considerably. Regarding the PGPR, the genera *Azospirillum*, *Bacillus*, and *Pseudomonas* showed positive effects for this moderator variable, increasing the number of nodules in 11.05% (95% CI [1.90–19.48%]), 26.05% (95% CI [14.71–36.59%]), and 10.41% (95% CI [3.43–17.41]), respectively (Fig. 4B). In relation to

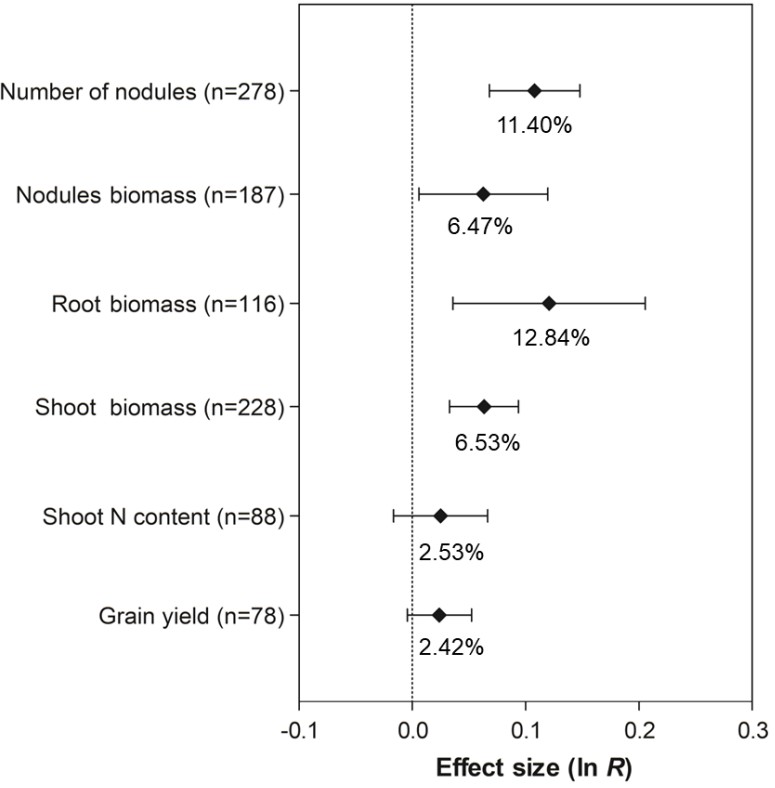

**Figure 3** **Effect sizes (ln *R*) of PGPR co-inoculation on nodule numbers, nodule biomass, root biomass, shoot biomass, shoot N content and grain yield.** The graph reflects the parameter estimates from the random-effects meta-analysis model conducted for each variable, and the error bars represent the 95% confidence interval. The values below the effect size of each variable are the percentages of the PGPR effect (ln *R* transformed back to the original values).

PGPR, only the genus *Bacillus* presented significant effects, leading to average increments of 33.12% (95% CI [22.27–44.93%]) (Fig. 4C). In contrast, in the pot experiments, the genera *Azospirillum*, *Bacillus*, and *Pseudomonas* presented significant effects of 26.77% (95% CI [8.26–48.44]), 22.09% (95% CI [6.67–39.72%]), and 9.81% (95% CI [2.13–26.30%]) on the number of nodules, respectively (Fig. 4D).

As shown in Fig. 5A, only the experiments conducted in pots showed significant effects on nodule biomass, with an average increase of 9.50% (95% CI [1.40–18.40%]). As for PGPR, the genera *Azospirillum* and *Pseudomonas* presented positive effects on this trait, showing increases of 14.65% (95% CI [6.76–23.13%]) and 17.34% (95% CI [7.17–29.49]), respectively (Fig. 5B). Although no significant effect of co-inoculation on nodule biomass was observed in the experiments conducted under field conditions, the partitioning of this effect in relation to the PGPR genera indicated a positive and significant effect of the genus *Azospirillum*, increasing the value of the trait in 10.69% (95% CI [3.70–18.16]) (Fig. 5C). In contrast, different PGPR in the pot studies revealed that only the genus *Pseudomonas* showed significant improvements in nodule biomass, presenting an increase of 16.80% (95% CI [6.58–27.90]) (Fig. 5D). On the other hand, a reduction of −18.32% in the

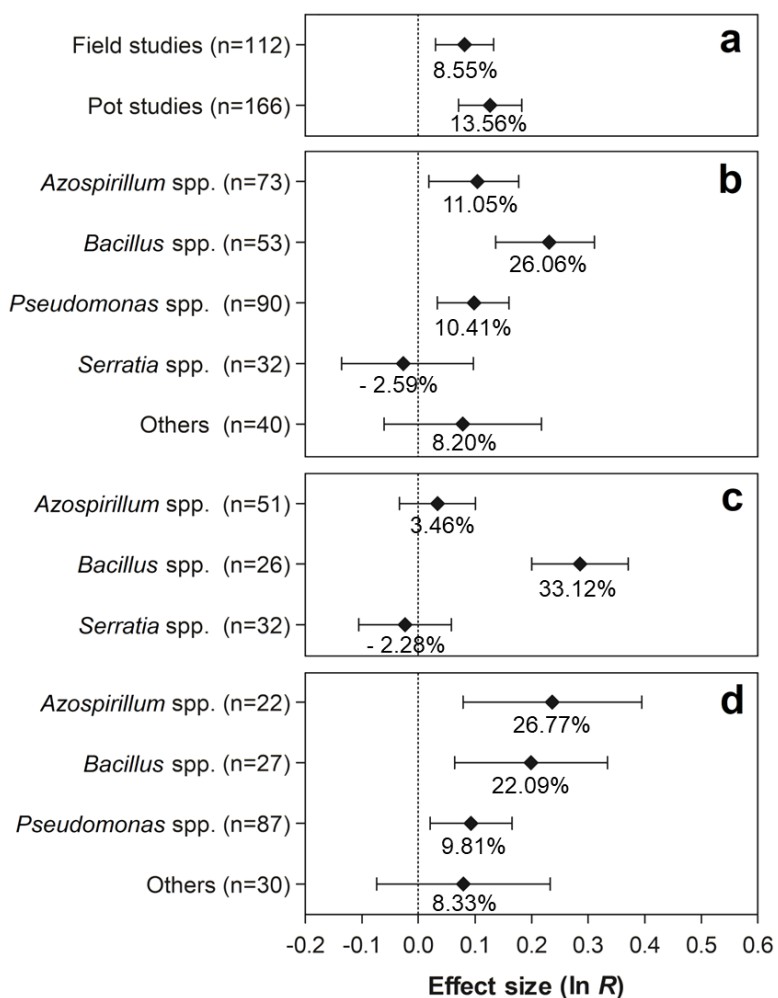

**Figure 4** **Effect sizes (ln R) of PGPR co-inoculation on number of nodules grouped by the moderator variables: (A) experimental conditions; (B) genera of PGPR; (C) genera of PGPR under field conditions; and (D) genera of PGPR under pot conditions.** The graph reflects the parameter estimates from the random-effects meta-analysis model conducted for each variable, and the error bars represent the 95% confidence interval. The values below the effect size of each variable are the percentages of the PGPR effect (ln R transformed back to the original values).

average nodule biomass (95% CI [−32.08–1.74]) was observed by co-inoculation of other PGPR genera (*Actinomadura*, *Aeromonas*, *Bacillus*, *Enterobacter*, *Herbaspirillum*, *Nocardia*, *Nonomuraea*, *Pseudonocardia*, *Rhizobium*, and *Streptomyces*).

The effects of the moderator variables on root biomass are presented in Fig. 6. For the experimental conditions, only the experiments conducted in pots showed significant values, with an increase of 15.79% (95% CI [4.33–28.49%]) in root biomass (Fig. 6A). Regarding PGPR, the genus *Pseudomonas* was the only one with a positive effect on this trait, presenting an increment of 28.89% (95% CI [10.93–49.77%]) (Fig. 6B). Furthermore, according to the results, only the genus *Pseudomonas* resulted in a significantly increased root biomass (28.96%) (95% CI [10.68–50.25%]) (Fig. 6C).

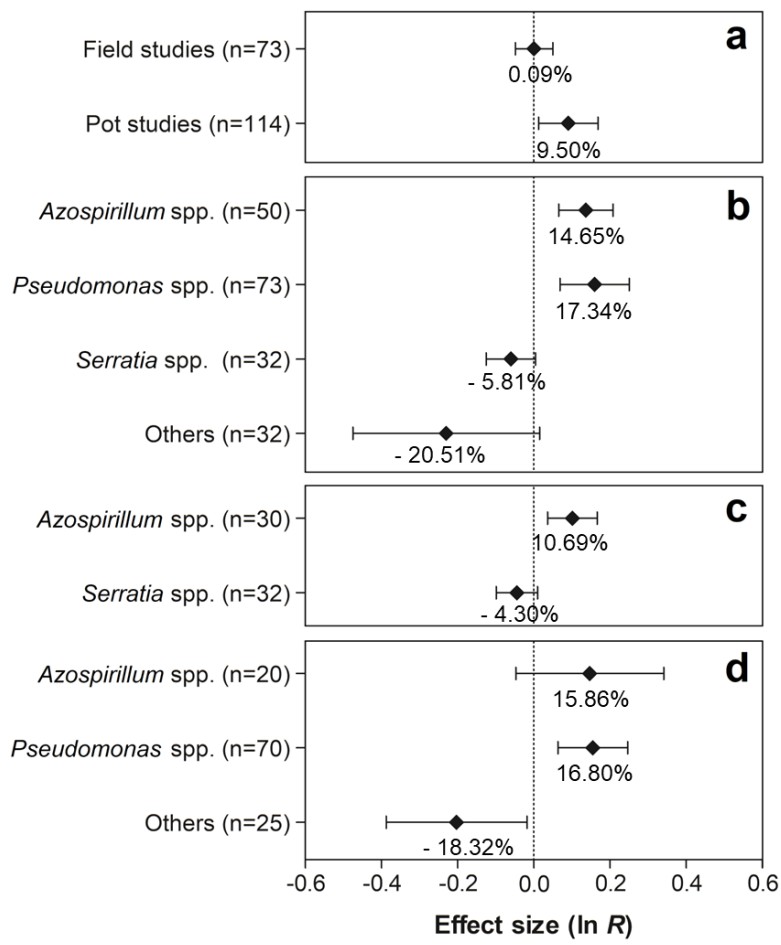

**Figure 5** Effect sizes (ln *R*) of PGPR co-inoculation on nodule biomass grouped by the moderator variables: (A) experimental conditions; (B) genera of PGPR; (C) genera of PGPR under field conditions; and (D) genera of PGPR under pot conditions. The graph reflects the parameter estimates from the random-effects meta-analysis model and the error bars represents the 95% confidence interval. The values below the effect size of each variable are the percentages of the PGPR effect (ln *R* transformed back to the original values).

Figure 7 shows the effects of the moderator variables on the shoot biomass. When the experimental conditions were analyzed, it was possible to verify that the trials carried out under field and pot conditions presented significant values of 5.44% (95% CI [3.14–7.80%]) and 8.27% (95% CI [3.06–13.76%]), respectively (Fig. 7A). Both effect sizes can be considered similar, since the IC overlapped considerably. For this moderate variable, the genera *Azospirillum*, *Bacillus*, and others (*Actinomadura*, *Aeromonas*, *Enterobacter*, *Herbaspirillum*, *Methylobacterium*, *Nocardia*, *Nonomurae*, *Pseudocardia*, *Rhizobium*, *Stenotrophomonas*, and *Streptomyces*) were the only ones that presented positive effects on shoot biomass, leading to increases of 6.39% (95% CI [3.12–9.76%]), 4.92% (95% CI [1.82–8.12%]), and 31.46% (95% CI [22.07–41.58]), respectively (Fig. 7B). The partitioning of PGPR genera under field conditions indicated that co-inoculation with bacteria of the genus

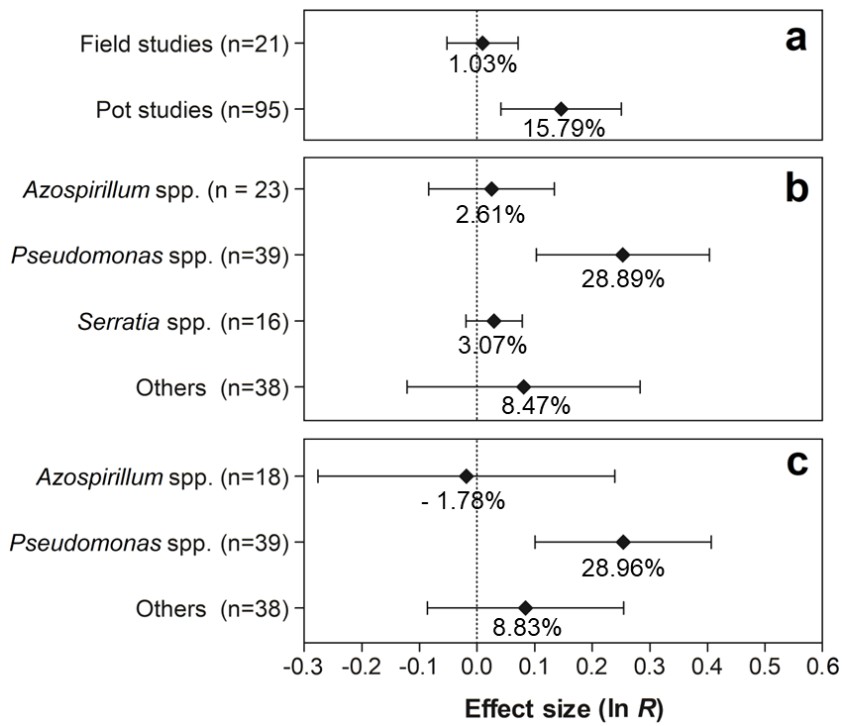

**Figure 6** Effect sizes (ln *R*) of PGPR co-inoculation on root biomass grouped by the moderator variables: **(A)** experimental conditions; **(B)** genera of PGPR; and **(C)** genera of PGPR under pot conditions. The graph reflects the estimates of the effects of the parameter estimates from the random-effects meta-analysis model and the error bars represent the 95% confidence interval. The values below the effect size of each variable are the percentages of the PGPR effect (ln *R* transformed back to the original values).

*Azospirillum* increased plant biomass in 5.42% (95% CI [2.95–7.95%]) (Fig. 7C). In the pot trials, an extra 28.39% (95% CI [17.50–40.27%]) in the average shoot biomass (Fig. 7D) was promoted by the grouped genera (*Actinomadura, Aerobonas, Enterobacter, Herbaspirillum, Methylobacterium, Nocardia, Nonomurae, Pseudocardia, Rhizobium, Stenotrophomonas*, and *Streptomyces*).

For the traits shoot N content and grain yield, none of the differences were statistically significant, since the 95% CI of the moderator variables overlapped with zero (Figs. 8 and 9).

## DISCUSSION

The soybean co-inoculation technology, in which traditional inoculation with selected strains of *Bradyrhizobium* is enhanced by the addition of bacteria considered plant growth promotors, has shown prominent results due to the complementary effects that these additional microorganisms promote. Whilst *Bradyrhizobium* acts as a microsymbiont, colonizing the plant root system and inducing the formation of nodules, PGPR increase root volume and number, thus enhancing the action of *Bradyrhizobium* in the supply of N biologically fixed to the plant, thereby potentially increasing grain yield (*Hungria,*

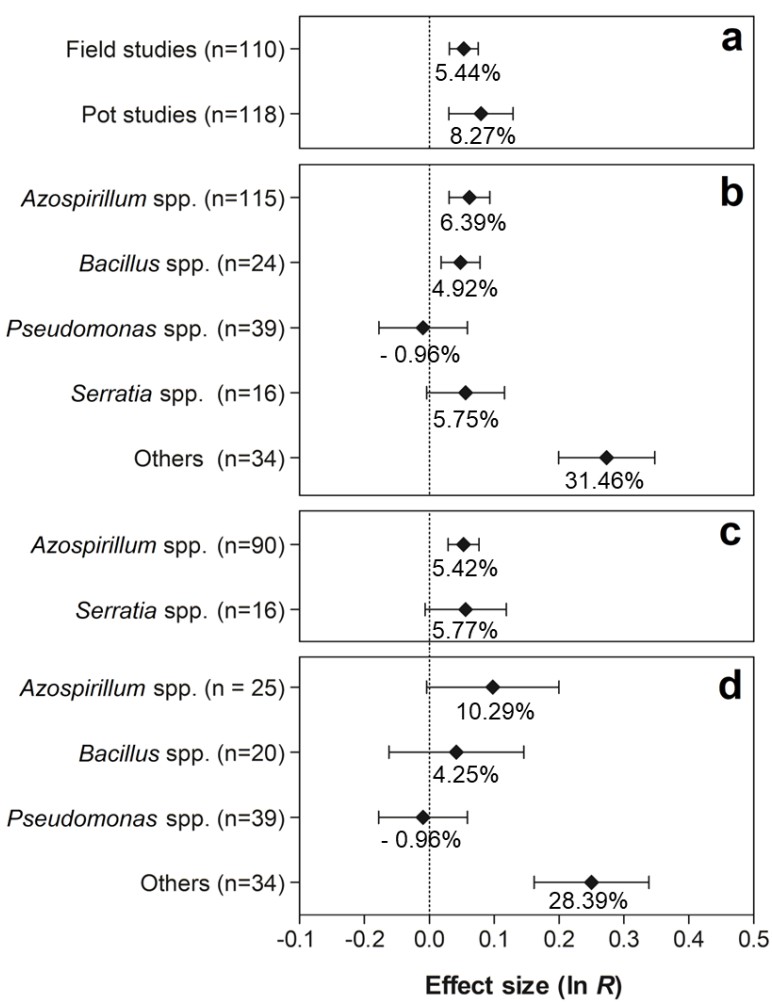

**Figure 7** **Effect sizes (ln $R$) of PGPR co-inoculation on shoot biomass grouped by the moderator variables: (A) experimental conditions; (B) genera of PGPR; (C) genera of PGPR under field conditions; and (D) genera of PGPR under pot conditions.** The graph reflects the parameter estimates from the random-effects meta-analysis model and the error bars represent the 95% confidence interval. The values below the effect size of each variable are the percentages of the PGPR effect (ln $R$ transformed back to the original values).

*Nogueira & Araujo, 2013*; *Hungria, Nogueira & Araujo, 2015*). However, the literature lacks a quantitative synthesis of the real contribution of the co-inoculation technology to the soybean crop. Therefore, the results obtained in the present meta-analysis have great relevance for our understanding of the responses to the co-inoculation of symbiotic and associative bacteria in soybean cultivation, with implications for the commercialization of PGPR-based mixed inoculants.

Co-inoculation of soybean with PGPR provides increments in traits of great importance for obtaining high grain yields, such as number of nodules as well as nodule, root, and shoot biomass. Previous studies have demonstrated the existence of positive correlations between these traits and grain yield, although the interaction effects of genotype-genotype

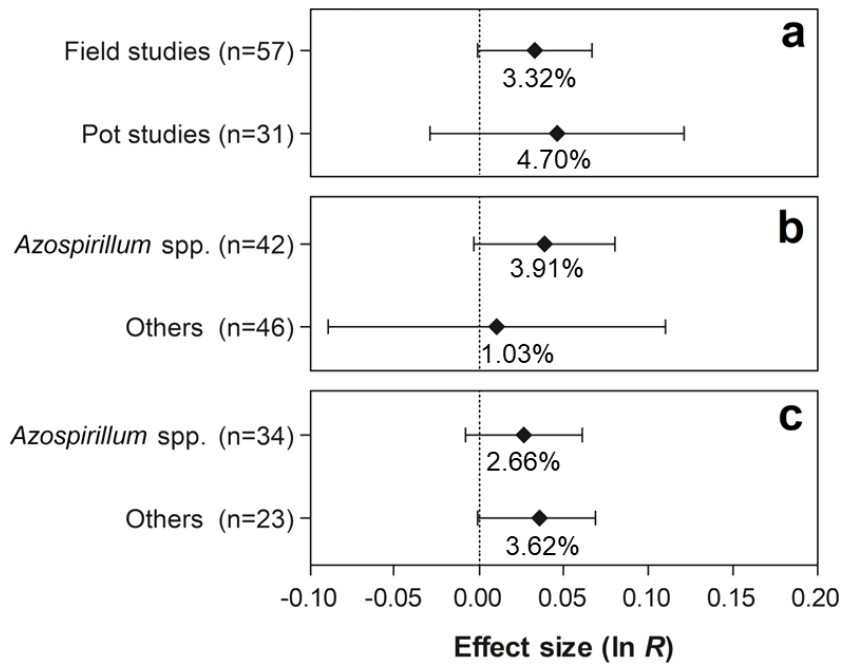

**Figure 8** Effect sizes (ln *R*) of PGPR co-inoculation on the shoot N content grouped by the moderator variables: **(A)** experimental conditions; **(B)** genera of PGPR; and **(C)** genera of PGPR under field conditions. The graph reflects the parameter estimates from the random-effects meta-analysis model and the error bars represent the 95% confidence interval. The values below the effect size of each variable are the percentages of the PGPR effect (ln *R* transformed back to the original values).

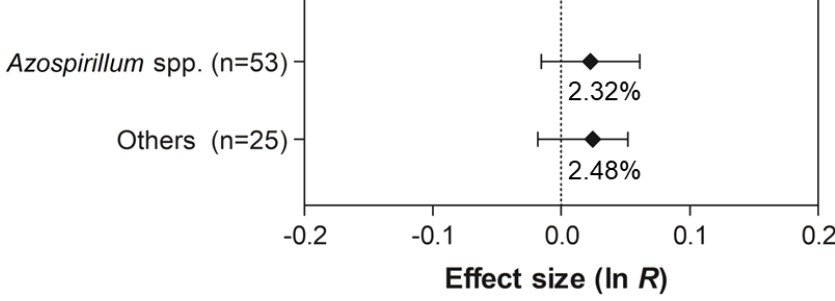

**Figure 9** Effect sizes (ln *R*) of PGPR co-inoculation on grain yield considering the PGPR genera moderator variable. The graph reflects the parameter estimates from the random-effects meta-analysis model and the error bars represent the 95% confidence interval. The values below the effect size of each variable are the percentages of the PGPR effect (ln *R* transformed back to the original values).

(macrosymbiont-microsymbiont) and genotype-environment are highlighted (*Hwang et al., 2014*; *Cui et al., 2016*; *Thilakarathna & Raizada, 2017*).

Meta-analysis studies quantifying the effects of PGPR on promoting plant-growth in different agricultural crops have been reported previously. *Veresoglou & Menexes (2010)* observed a significant increase of 23.81% in shoot biomass of wheat (*Triticum aestivum* L.)
when inoculated with *Azospirillum* spp. Corroborating results were found by *Rubin, Van Groenigen & Hungate (2017)*, who reported higher shoot and root biomass production (28 and 35%, respectively) induced by PGPR in a range of plant species. Furthermore, verifying the influence of inoculation with *Azospirillum* spp. in maize, interesting results were found by *Zeffa et al. (2018)*, where the inoculated treatment out-yielded the control by 651 kg ha$^{-1}$. In general, it is believed that the production of phytohormones by PGPR is one of the main mechanisms of action on the development of the host plant, whose effects are more prominent on the root system (*Olanrewaju, Glick & Babalola, 2017*; *Puente et al., 2018*). Interestingly, the symbiotic relationship between rhizobia and legumes is also mediated by bacterial phytohormones (*Stacey et al., 1995*; *Imada et al., 2017*). In this context, auxins produced by PGPR are believed to increase the number of root hairs, leading to the formation of rhizobia-soybean interaction sites (*Schmidt, Messmer & Wilbois, 2015*).

*Puente et al. (2018)* examined the effect of IAA on the co-inoculation response of soybean with *Bradyrhizobium* and *A. brasilense* and demonstrated that the increase in root system growth, which improves the soybean-*Bradyrhizobium* interaction, is a result of the action of phytohormones. Moreover, the authors co-inoculated soybean with *A. brasilense* Az39 (*ipdC*+) and with its respective mutant deficient in IAA biosynthesis (*ipdC*-). The authors observed that co-inoculation with *A. brasilense* Az39 promoted a greater efficiency in the *Bradyrhizobium*-soybean symbiosis when compared to the treatment of co-inoculation with the mutant (Az39 *ipdC*-) or the application of synthetic IAA and concluded that both the presence of *Azospirillum* and IAA biosynthesis by these bacteria are responsible for the positive effects of soybean co-inoculation with *Bradyrhizobium* and PGPR. Several other studies have linked phytohormone production to the successful interaction between rhizobia and legumes (*Fukuhara et al., 1994*; *Srinivasan, Holl & Petersen, 1996*; *Vicario et al., 2015*).

Although the correlation between nodulation parameters in soybean (nodule number and nodule biomass) is already widely described, the data assembled by the present meta-analysis indicated no significant increase in grain yield and shoot N content as a result of soybean co-inoculation compared to conventional inoculation (only *Bradyrhizobium*). It is important to emphasize that the meta-analysis for grain yield considered only data from field studies, in which the variables are difficult to control, such as the presence of native strains competing with the inoculant for nodulation. Furthermore, soybean responses to co-inoculation may vary according to plant genotype, bacterial strain, environmental conditions, as well as the quantity and quality of PGPR cells used as inoculants (*Schmidt, Messmer & Wilbois, 2015*; *Pannecoucque et al., 2018*; *Chibeba et al., 2018*). These variations in responses to co-inoculation were evident in the studies evaluated, which can be observed in the CI for different PGPR strains, in all the traits described.

The results of this meta-analysis point to a lack of a positive and significant contribution of co-inoculation to soybean grain yield. Nevertheless, indirect evidence indicates that the identification of inoculant strains that cause complementary effects on plant development is a crucial step for the development of more efficient soybean inoculants. Moreover, based on the analysis of the data gathered, it can be concluded that the improvement of soybean tolerance to abiotic stresses (such as drought and high temperatures) can be achieved by

co-inoculation, since significant increases have been demonstrated for plant biomass and nodule number and biomass when this technique was applied.

In general, the results obtained in the present meta-analysis indicate the need for more experimental data from field experiments to produce more robust analyses to assess the real contribution of the co-inoculation technology for soybean cultivation. Among the traits that did not present statistical significance, shoot N content and grain yield were the ones with the lowest numbers of observations considered in the analysis. This situation is reinforced by the fact that co-inoculation of soybean with PGPR is more effective for experiments in pots compared to experiments conducted in the field. In addition to greater environmental control, the reader should bear in mind that experiments in pots present a less diverse native bacterial community compared to native soils, which means a greater competition between inoculant organisms and soil bacterial communities in field experiments (*Çakmakçi et al., 2006*).

## CONCLUSIONS

Our results demonstrated that the co-inoculation of soybean with *Bradyrhizobium* and other PGPR can substantially increase nodule number (11.40%), nodule biomass (6.47%), root biomass (12.84%), and shoot biomass (6.53%) in soybean. On the other hand, no significant differences were observed for shoot N content and grain yield. The bacterial genera *Azospirillum*, *Bacillus*, and *Pseudomonas* were more effective when compared to the genus *Serratia*. In general, co-inoculation results were more pronounced in experiments conducted in pots than in the field. The co-inoculation technology can be considered an economically viable and environmentally sustainable strategy for soybean cultivation.

### Funding
This study was financed by the Coordenação de Aperfeiçoamento de Pessoal de Nível Superior–Brasil (CAPES)–Finance Code 001. The funders had no role in study design, data collection and analysis, decision to publish, or preparation of the manuscript.

### Grant Disclosures
The following grant information was disclosed by the authors:
Coordenação de Aperfeiçoamento de Pessoal de Nível Superior–Brasil (CAPES)–Finance Code 001.

### Competing Interests
The authors declare there are no competing interests.

### Author Contributions
- Douglas M. Zeffa conceived and designed the experiments, performed the experiments, analyzed the data, contributed reagents/materials/analysis tools, prepared figures and/or tables, authored or reviewed drafts of the paper, approved the final draft, search Strategy.
- Lucas H. Fantin conceived and designed the experiments, performed the experiments, analyzed the data, contributed reagents/materials/analysis tools, prepared figures and/or tables, authored or reviewed drafts of the paper, approved the final draft.
- Alessandra Koltun performed the experiments, prepared figures and/or tables, authored or reviewed drafts of the paper, approved the final draft.
- André L.M. de Oliveira, Marcelo G. Canteri and Leandro S.A. Gonçalves conceived and designed the experiments, performed the experiments, authored or reviewed drafts of the paper, approved the final draft.
- Maria P.B.A. Nunes performed the experiments, analyzed the data, prepared figures and/or tables, authored or reviewed drafts of the paper, approved the final draft.

## Data Availability

    The raw measurements are available in the Supplemental Files.

## Supplemental Information

Supplemental information for this article can be found online at http://dx.doi.org/10.7717/peerj.7905#supplemental-information.

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
