# Peer review of "Effects of plant growth-promoting rhizobacteria on co-inoculation with Bradyrhizobium in soybean crop: a meta-analysis of studies from 1987 to 2018"

_PeerJ, doi:10.7717/peerj.7905_

## Round 0.1 · original submission · Minor Revisions

Dear Dr. Leandro Gonçalves,

I have received the comments of three independent reviewers regarding your MS “Effects of plant growth-promoting rhizobacteria on co-inoculation with Bradyrhizobium in soybean crop: a meta-analysis of studies from 1987 to 2018”. Based on that, I suggest that you address the minor revisions according to the reviewer’s suggestions. Please pay particular attention in the requested detailed information that is missing (particularly in the Introduction and References) as well as on the development of key points in the Discussion and Conclusions. The figures legends should also be improved.


Sincerely,
Ana I. Ribeiro-Barros

·

Basic reporting

Overall this was a useful study. It integrates a wide range of data already present in the peer reviewed literature. This kind of integration has clear value, although it can sometimes overlook important details, such as specific conditions where the inputs under investigation cause yield increases.

Experimental design

The papers on which the research was based were all peer reviewed and so very likely to have been based on research with appropriate experimental designs. As to the meta-analysis, we do not have expertise in this area and so cannot attest to the appropriateness of this aspect of the work.

Validity of the findings

The findings, overall, appear to be valid and, from an agronomic/biological perspective, not entirely surprising. Again, we worry that there might have been some important details that have been buried in the work. However, that is not necessarily something that needs to be specifically addressed in this paper, although it would be good to provide some small comment/acknowledgement of this situation.

Additional comments

General comments:
1. Overall: It is very useful to have a meta-analysis across a wide range of published research data. This provides a strong overall view of the utility of specific inputs. However, it may be that an input is effective under specific conditions, and this could be lost in such an analysis.
2. Line 21: This does seem to state the main problem; the authors give a quantification of other effects on the soybean crop, and yield, only for the field crops, and the reasons for this are clear. In the end, there were no significant yield effects under overall field conditions. Would there be yield effects under specific field conditions or with specific sets of microbe additions?
3. Introduction: It might be good to add at least a brief comment on the environmental and economic benefits of this symbiosis.
4. Introduction: A very short comment of the power and specific utility of meta-analysis would be useful.
5. Introduction/Discussion: Some of the material in the Discussion is more of a background nature and could probably be moved to the Introduction. This material could be replaced with a bit more consideration around the agronomic implications of the findings.
6. Discussion: It might be useful to have a bit more on the meta-analysis approach and how this one compares.
7. Line 240: Is this because more nodules means more nitrogen fixation?
8. Lines 282-284: This is the main agronomic finding. Could some additional consideration regarding specific conditions or treatments be given. If some treatments or conditions result in important improvements in yield, could this be indicated with at least a very rudimentary consideration of economic benefit?
9. Lines 318-319: Perhaps this should be qualified, given the lack of yield effect. Could some specific types of treatments be more promising? Would such treatments, given the low cost of these kinds of inputs, be economically viable?
10. Overall: From a biological perspective the pot studies may be of greater interest, while from an agronomics point of view, the fieldwork is clearly the most important part. Were there key differences between the findings of two (aside from the obvious inability to collect yield data in the controlled environment work)?
11. Language: In general the use of language was quite good. However, we have marked some instances, on an accompanying PDF of the manuscript, that could be improved.

Reviewer 2 ·

Basic reporting

The present study was aimed to provide a quantification of the effects of the co-inoculation of Bradyrhizobium and PGPR on the soybean crop using a meta-analysis approach.
The manuscript is well designed and written, so, I would like to support this manuscript to publish in PeerJ, with recommended MINOR revision. The co-inoculation of soybean with Bradyrhizobium and other plant growth-promoting rhizobacteria (PGPR) is considered a promising technology. However, there has been little quantitative analysis of the effects of this technique on yield parameters.

Experimental design

The bibliographic research and data collection was well planned and performed and it collects a series of measures on the different level of plant attributes. The experimental design was appropriate, and the Meta analysis was applied correctly. Authors used very progressive methods and protocols, and a total of 42 published articles were examined.

Validity of the findings

Overall, this manuscript represents good and new findings, however, some suggestions should be considered before publication.

Additional comments

Dear editor in chief of the PeerJ,

The present study was aimed to provide a quantification of the effects of the co-inoculation of Bradyrhizobium and PGPR on the soybean crop using a meta-analysis approach.
The manuscript is well designed and written, so, I would like to support this manuscript to publish in PeerJ, with recommended MINOR revision.
The co-inoculation of soybean with Bradyrhizobium and other plant growth-promoting rhizobacteria (PGPR) is considered a promising technology. However, there has been little quantitative analysis of the effects of this technique on yield parameters.
The topic of the research is coherent to the aims of the PeerJ. Abstract is clear, informative and precise. The introduction is very clearly stated and comprehensive. The bibliographic research and data collection was well planned and performed and it collects a series of measures on the different level of plant attributes. The experimental design was appropriate, and the Meta analysis was applied correctly. Authors used very progressive methods and protocols, and a total of 42 published articles were examined.
Overall, this manuscript represents good and new findings, however, some suggestions should be considered before publication as follows:
-Line 35, "…were more pronounced under pot than under field conditions" should be changed to "… were more pronounced under pot than that of field conditions".
Line 36, "these studies outline that" should be changed to "this study summarize that"-
- Line 76-77, please check the writing style of "Hungria et al. (Hungria, 77 Nogueira & Araujo, 2013)"
- In materials and methods section (line 106), the authors should clarify why interaction data with biotic or abiotic stresses were not extracted from articles?
- Line 129, please include the reference of equation.
- The sentence in Line 232 (since the 95% CI of the moderator variables overlapped with zero) is not clear, please explain it precisely.
- Authors could add the new following references to develop discussion about coinoculation of bioinoculants to improve plant performance:
- Coinoculation of bioinoculants improve Acacia auriculiformis seedling growth and quality in a tropical Alfisol soil. J. For. Res., 2018, 29(3): 663-673.
- Leaf nitrogen and phosphorus resorption of woody species in response to climatic conditions and soil nutrients: a meta-analysis. J. For. Res., 2018, 29(4): 905-913.
- The description of the illustrations (in Figures) is insufficient, no explained all symbols and abbreviations used.
- The conclusion must be more defined without unnecessary discussion.

Reviewer 3 ·

Basic reporting

I think the ms is well-written and the results are mostly supported by the analyses; however, I have some concerns, stated below.
The language is clear, and the English style is fine, although I do not feel qualified to judge it.

Experimental design

The approach taken by the authors represents nowadays a very reliable way to compare and analyse a plethora of studies that sometimes have contradictions. This tool may be helpful for the scientific community and in my opinion, it should be more used, even in the elaboration of introduction sections and or other sections of actual manuscripts.
There are some concerns I would like to state here:

1.- Even the authors provide explanation in lines 131-136, I am not sure whether is good to compare pots vs field assays, even if you chose that meta-analyses approach. Did the authors of the meta-analysed papers involving pot assays reached production?

2.- The authors said that they compared a final amount of 42 papers from 1987 to 2018; surprisingly, in the supplementary data S1, there are only 41 papers….is this a mistake? Moreover, some of these papers only contains 2 comparable data (Teixeira Filho et al 2017 and Galindo et al 2018). I was wondering if it is a good idea to include them or not in the study.

3.-One factor that was not contemplated by the authors is the inoculation method, regarding to how the inocula was added to the plant/soil (liquid, solid, with/without carrier, pelleted seeds…). I think that this is one of the key points in the success of an inoculum.

4.- It will be nice to read a little bit more discussion about the reasons of why the inoculated plants did not show significant increases in yield/production. This fact leads to the question of the success of isolates that are great in controlled conditions and then in soils are not, due to a bunch of reasons. I am aware that the authors include a paragraph (301-310) discussing some of the reasons, but I think they should develop the topic further.

More comments:

Lines 53-55. Valid species of the genus Bradyrhizobium are published in official journal, such as IJSEM, and/or validated in the Validation list. NCBI is not the best database for retrieving data on bacterial species. Please, check IJSEM or LSPN database. Also, in line 53, please add the complete reference of who described first the genus Bradyrhizobium

Lines 75-88. This paragraph is about the co-inoculation of Bradyrhizobium and Azospirillum; however, a co-inoculation with B. subtilis appears somehow. It is not clear if this is a mistake or the authors wanted to explain other co-inoculation assay.

Line 100-101 If the main criteria is that the evaluated papers should be written in English, Portuguese or Spanish, please, add this information in the supplementary data S1. Even I recognise there might be excellent works written in Portuguese and Spanish, I think only English-written papers should be evaluated.

Validity of the findings

The manuscript presented by Zeffa et al showed that meta-analyses approach might be an excellent approach for comparing really diverse data form many publications. Conclusions are well supported by the results of the analyses. I just have some comments that make me recommend to authors performing Minor Revisions.

---

## Round 0.2 · accepted · Accept

Dear Dr. Gonçalves,

I have received the revised version of your MS, which incorporates nearly all the suggestions made by the reviewers. Thus, I would like to inform you that the paper is now accepted for publication.

Thank you for choosing PeerJ to publish your work and for keeping the quality standards of the journal,

Sincerely,
Ana I. Ribeiro-Barros